# DEFENDING AGAINST MODEL EXTRACTION FOR GNNs WITH MODEL REPROGRAMMING

## ABSTRACT

The goal of model extraction (ME) on Graph Neural Networks (GNNs) is to steal the functionality of GNN models. Defense against extracting GNN models faces several challenges: (1) Existing defense primarily designed for defense against convolutional neural networks without considering the graph structure of GNNs; (2) Watermarked-based defense is typically passive without preventing model extraction from happening and can only identify a model stealing after extraction has occurred; (3) They either require entirely defensive training from scratch or expensive computation during inference. To address these limitations, we propose an effective defense method that can reprogram the model with graph structure-based and layer-wise noise to prevent ME for GNNs while maintaining model utility. Specifically, we reprogram the target model to: (1) introduce graph structure-based disturbances that prevent the attacker from fully learning its functionality; (2) incorporate data-specific, layer-wise noise into the target model to enhance defense while maintaining utility. Therefore, we can prevent the attacker from extracting the reprogrammed target model and preserve the model's utility with improved inference efficiency. Extensive experiments and analysis on defending against both hard-label and soft-label ME for GNNs demonstrate that our strategy can lessen the effectiveness of existing attack strategies while maintaining the model utility of the target model for benign queries.

## 1 INTRODUCTION

In recent years, Graph Neural Networks (GNNs) Kipf & Welling (2016); Hamilton et al. (2017); Veličković et al. (2017) have been heavily used in critical domains, including the API services. Pretrained graph models are also used with third parties for various downstream tasks Liu et al. (2023); Long et al. (2022). However, GNNs are vulnerable to ME attacks. ME attacks on GNNs DeFazio & Ramesh (2019); Wu et al. (2022a); Shen et al. (2022); Zhuang et al. (2024) involve an adversary attempting to replicate a target GNN's functionality by systematically querying it and using the responses to train a surrogate model. GNNs, designed to operate on graph-structured data, pose unique challenges and opportunities for attackers due to their complex architectures and interdependencies among nodes and edges. Many web services, including recommendation systems Wu et al. (2022b), fraud detection platforms Liu et al. (2021), and social media Li et al. (2023), rely on graph data and GNNs for efficient and accurate predictions. Consequently, defending against model extraction attacks on GNNs is critical for maintaining both security and intellectual property.

While prior work has made progress in defending against model extraction attacks, there remains a significant gap: most defenses are either reactive or computationally demanding, and few consider the specific properties of graph-structured data. This raises a central research question:

> *Q: How can we design an active defense for GNNs that prevents model extraction while maintaining high utility for benign queries, leveraging graph structure and layer-wise noise?*

In this work, we address a specific scientific challenge in the web domain, focusing on the vulnerability of GNNs used in web-based applications and protecting the intellectual property of GNNs.

There are numerous studies on defense against model extraction attacks. Current research of model extraction defense methods generally falls into the following categories Wang et al. (2023): (1)

*Passive defenses*: they aim to monitor and detect ongoing extraction attempts Juuti et al. (2019); Pal et al. (2021) or verify whether the target model has been stolen Jia et al. (2021); Szyller et al. (2021); Maini et al. (2021). These methods focus on observing query patterns and identifying unusual activities that may include extraction behaviors. Nevertheless, it often fails to prevent adaptive attackers who can analyze and circumvent these defenses over time; (2) *Active defenses*: they aim to prevent model extraction attacks from happening Orekondy et al. (2019b); Kariyappa & Qureshi (2020); Kariyappa et al. (2021b); Mazeika et al. (2022); Wang et al. (2023). These methods focus on maximally reducing the accuracy of the clone model.

In this work, our method belongs to the category of active defense. Existing defenses are either computationally intensive, requiring costly test-time optimization that limits their practicality Orekondy et al. (2019b); Mazeika et al. (2022), or memory intensive, increasing deployment complexity Kariyappa et al. (2021b). Additionally, existing defense methods are primarily designed for image data using CNNs and fail to account for graph-structured information.

Model Reprogramming Chen (2024); Jing et al. (2023) typically involves adapting a pre-trained model to perform a different task without modifying its underlying architecture or weights. Inspired by model reprogramming, we reprogram the pre-trained target model to perform correct predictions for legitimate users but produce misleading or less informative results for adversarial queries. In this work, we propose an effective defense method that can reprogram the model with graph structure-based and layer-wise noise to prevent ME for GNNs while maintaining model utility. Specifically, we design learnable layer-wise noises into the hidden layers in the GNNs that can maintain utility on in-distribution benign queries and decrease the performance on out-of-distribution attack queries. Besides, we apply the graph-structure features to modify the layer-wise noise range to enhance the defense ability to prevent the attacker from stealing the functionality of the victim model.

In summary, our main contributions are three-fold:

- We propose an efficient active defense framework through model reprogramming to efficiently defend against model extraction on GNNs without needing full retraining or expensive computation during testing.
- We propose to leverage the graph structure of GNNs to further enhance our defense effectiveness.
- We provide detailed theoretical analysis and support for our proposed defense, guaranteeing the performance of our defense method.
- Extensive experiments on defending against model extraction for GNNs show that our method can maintain high model utility for benign attacks and reduce inference time cost.

## 2 RELATED WORKS

### 2.1 MODEL EXTRACTION ATTACK

**Model Extraction** (ME), also known as model stealing, refers to the process of replicating a target machine learning model by either extracting its parameters or approximating its functional utility Orekondy et al. (2019a); Papernot et al. (2017); Truong et al. (2021); Oliynyk et al. (2023).

The attacker's goal in ME is to create a clone model that performs similarly to the target model without having direct access to its internal structure or training data. The objective for extracting a model from MLaaS system is multiple, including stealing exact model such as the learned parameters Reith et al. (2019), training hyperparameters Wang & Gong (2018), or the architecture of the target model Oh et al. (2019). Besides, the objective can also be stealing model behavior, including copying the target model that reaches the same level of effectiveness and approximating the behavior of producing the same outputs with the target model. In this research, we will focus on the attack that recovers the same level of functionality of the target model with a different architecture.

ME attacks can be broadly categorized into two main types: (1) **Data-Based Model Extraction (DBME)** Kariyappa et al. (2021a); Correia-Silva et al. (2018); Papernot et al. (2017) involves an attacker querying the target model with a set of input data to gather corresponding output responses. Using these input-output pairs, the attacker trains a clone model that approximates the behavior of the target model. This method relies on the availability of data and the ability to query the model multiple times. (2) **Data-Free Model Extraction (DFME)** Kariyappa et al. (2021a); Truong et al. (2021);

Wang et al. (2023) involves an attacker extracting the target model's functionality with artificial data rather than relying on any real input data. DFME is particularly challenging because it does not assume access to the original training data or similar datasets, making it a more sophisticated attack method.

ME attacks can also be grouped in to two categories based on the query outputs of the MLaaS: (1) The **soft-label** ME will only evaluate the probability logits difference between the output of target model and clone model under in-distribution settings. (2) The **hard-label** ME will have less information on output for queries, it only provides the class with the maximum probability in classification tasks. Besides, it will be much more challenging to train the reprogrammed target model since the objective is different.

There are also a few recent works on model extraction attacks for GNNs DeFazio & Ramesh (2019); Wu et al. (2022a); Shen et al. (2022); Zhuang et al. (2024). Specifically, Wu et al. (2022a) applies discrete graph structure learning to construct a connected substitute graph using these node attributes; Shen et al. (2022) initializing the graph structure with kNN based on node attributes and updating it using a graph structure learning framework. Zhuang et al. (2024) is a data-free model extraction attack framework that can applied to several graph tasks.

## 2.2 MODEL EXTRACTION DEFENSE

Model extraction defenses can be broadly categorized into two classes: (1) **Passive Defenses**, which focus on monitoring and detecting extraction attempts without modifying the model's behavior Jia et al. (2021); Szyller et al. (2021); Maini et al. (2021). These methods analyze query patterns, distributions, frequency, and timing to identify suspicious activities. Standard techniques include logging and anomaly detection. (2) **Active Defenses**, which aim to prevent model extraction (ME) attacks by proactively modifying the model's responses or access mechanisms Orekondy et al. (2019b); Kariyappa & Qureshi (2020); Kariyappa et al. (2021b); Mazeika et al. (2022); Wang et al. (2023). Strategies include Prediction Obfuscation Orekondy et al. (2019b), which alters model outputs to reduce informativeness; Perturbation Techniques Kariyappa & Qureshi (2020); Kariyappa et al. (2021b); Wang et al. (2023), which introduce noise to hinder accurate extraction; and Query Limitation Mazeika et al. (2022), which restricts query rates or adds variability to responses. Additionally, Behavior-Based Defenses Wang et al. (2023); Zhuang et al. (2024) dynamically adjust responses based on user behavior analysis and query pattern detection to counter extraction attempts effectively.

Model extraction defense on GNNs can be challenging since the graph data can include node features, edge features, entity information, and structures; Current model extraction defense methods may not perform well on GNNs as they do not incorporate the structure of GNNs. Our work proposes a new pipeline to protect GNN stealing, emphasizing graph-structure sensitivity and adaptive layer-wise noise for the unique construction of Graph-based models. Our research based on model reprogramming will mainly focus on *active defense* since they can prevent ME before their occurrence and thus reduce clone probability. To our knowledge, this is the first study to explore model extraction defenses for GNNs.

## 2.3 MODEL REPROGRAMMING

Model reprogramming Chen (2024) enables the efficient reuse of pre-trained models across different tasks without changing original model parameters. This approach leverages the model's inherent generalization capabilities to handle new tasks by adjusting the input or output interpretation by reprogramming a model trained on a source domain for tasks in a target domain. This method provides a resource-efficient alternative to traditional transfer learning, making it suitable for scenarios where rapid deployment or adaptation to new tasks is needed. Model reprogramming can also be applied to graphs: Jing et al. (2023) allows reprogramming pre-trained GNN models to perform new tasks by adding learnable perturbations to the input graph data without modifying the original model architecture or parameter. By applying model reprogramming, we can adaptively fine-tune the model parameters without retraining, thereby reducing the time cost during the inference stage.

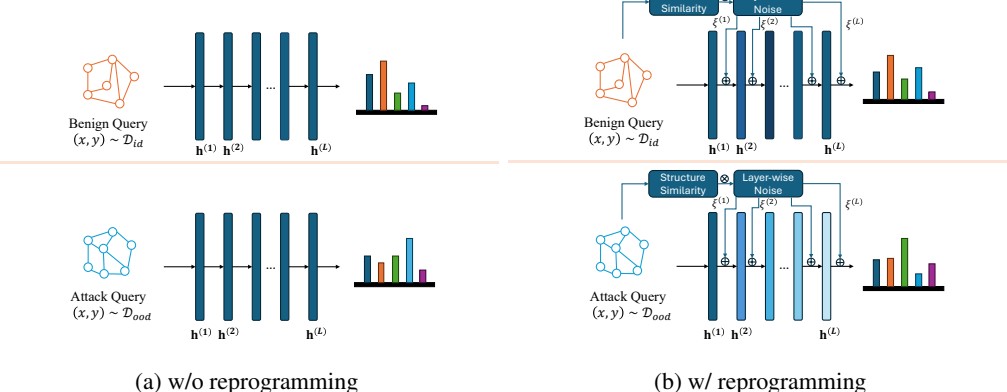

(a) w/o reprogramming          (b) w/ reprogramming

Figure 1: (a) The architectures without reprogramming for benign queries and attack queries. The benign input graph (with orange ground truth label) will receive the correct prediction, while the attack input graph (with blue ground truth label) will also get the correct prediction. (b) The architectures with reprogramming for benign queries and attack queries. The model undergoes reprogramming, transforming into different schemes with data-specific graph and structure features. This changes the prediction probabilities, leading to the benign query achieving a correct prediction (orange), while the attack query receives a misled prediction (green).

## 3 PRELIMINARY

### 3.1 DATASET

A graph $G$ is defined with node (vertices) set $V$ and edge set $E$, as $G = (V, E)$. For a graph with $n$ nodes, it can also be represented as $x = (A, X)$, where $A \in \mathbb{R}^{n \times n}$ is adjacency matrix and $X \in \mathbb{R}^{n \times d}$ is node feature, $d$ is dimension for graph features.

A dataset for graph classification tasks is given as $D = (x_i, y_i)_{i=1}^N$, sampled from a distribution $\mathcal{D}$, with the graph distribution being $\mathcal{G}$. We use $\mathcal{D}$ for the data distribution and use $D$ for datasets (data-label pairs) in this work.

In DBME defense tasks, the defender (target model) utilizes both in-distribution (private) data from $\mathcal{D}_{id}$ and out-of-distribution (public) data from $\mathcal{D}_{ood}$. Conversely, the attacker uses a model with a different architecture under mainly out-of-distribution data to clone the functionality of the target model. Under the settings of DFME for GNNs, the clone model can also use a graph generator to generate node features and an adjacency matrix. In practice, the dataset can be split into disjoint sets, $\mathcal{D} = \mathcal{D}_{id} \cup \mathcal{D}_{ood}$, where private data is sampled from $\mathcal{D}_{id}$, and synthetic or OOD data belongs to $\mathcal{D}_{ood}$. Similarly, for graph data, we have $\mathcal{G} = \mathcal{G}_{id} \cup \mathcal{G}_{ood}$.

### 3.2 MODEL

On the defender side, the target model (victim model) is defined as $\hat{y} = T(x; \theta_T)$, where $T$ maps the graph data $x$ to the class set $Y$. On the attacker side, the clone model is represented as $\hat{y} = C(x; \theta_C)$, with $C$ mapping out-of-distribution graph data $\mathcal{G}_{ood}$ to the class set $Y$. Additionally, the attacker uses a data generator $f_{gen}$ to generate graph data, mapping from an unspecified domain to $\mathcal{G}$.

### 3.3 ATTACKER'S GOAL AND KNOWLEDGE

The attacker's goal is to optimize the clone model parameters, denoted as $\theta_C$, by minimizing the divergence between the clone model $C(\cdot; \theta_C)$ and the target model $T(\cdot; \theta_T)$ over OOD query data from the output space $D_{ood}$. Specifically, the attacker minimizes the *Kullback-Leibler* (KL) divergence between the clone model's output and the target model's output:

$$\min_{\theta_C} \mathbb{E}_{(x,y) \sim \mathcal{D}_{ood}} D_{\text{KL}} \left[ C(x; \theta_C) \parallel T(x; \theta_T^*) \right] \tag{1}$$

where $\theta_T^*$ is the optimal parameter for the target model. The loss function guides clone model training, with KL divergence measuring how one probability distribution diverges from another. Minimizing

this divergence helps the attacker align the clone model's output with the target model's, capturing its behavior.

Following Wang (2021); Wang et al. (2023), the attacker can hardly access the original training data distribution of the target model; they can either use natural (DBME) or synthetic OOD data (DFME) from $\mathcal{D}_{ood}$ to query the target model to extract the functionality. Besides, the attacker does not know the architecture and model parameters of the target model, so the attacker is assumed to be trained and tested under OOD data with another model architecture. In the score-based settings (soft-label), the target model delivers all the probabilities of different classes to the clone model. In contrast, in the decision-based setting (hard-label), the attacker can only get the top-1 class prediction from the clone model.

The training objective of the clone model is to simulate the functionality of the target model, while the final goal of the attacker is to train a clone model $C$ with parameters $\theta_C$ that can reach high test accuracy on ID test data from $D_{id}^{test}$.

The defender assumes that the attacker's query data can be classified as out-of-distribution (OOD) data concerning the training dataset of the target model Wang (2021); Kariyappa et al. (2021a); Wang et al. (2023). This assumption is based on the premise that the attacker's inputs will differ significantly from the distribution of data the model was trained on.

### 3.4 DEFENDER'S GOAL AND KNOWLEDGE

The defender's primary goal is to maximize the test accuracy of the target model on the test ID data, which reduces the effectiveness of the attack. Suppose we use $l$ as the classification loss (e.g., cross-entropy loss $l_{CE}$)

$$\min_{\theta_T} \mathbb{E}_{(x,y)\sim\mathcal{D}_{id}}\left[l_{CE}(T(x,\theta_T), y)\right] \tag{2}$$

Besides, the defender's goals include minimizing the test accuracy that the attacker can achieve and preserving the utility. We also hope that the defense procedure will be memory-efficient and computation-efficient. In order to train the target model for better defense ability, we also apply an OOD dataset other than the ID dataset for training and evaluation.

To achieve this, the defender must proactively understand the attacker's behavior and adapt the target model accordingly. The defender possesses knowledge about the attacker's possible strategies, including the types of queries that may be used to extract information from the target model. This understanding allows the defender to anticipate potential vulnerabilities and implement countermeasures.

We have also discuss the assumption of considering the attack queries as as OOD data in section B.4.

## 4 METHOD

This section presents the architecture of our method in Figure 1b: **Graph Structure-Aware Layer-Wise Reprogramming**.

We present how our design will use graph data to defend against model extraction in a layer-wise manner in Section 4.2. We then discuss how graph structure will be applied in defensive learning in Section 4.3. We will illustrate the algorithm for the clone model (attacker) and the defensive training algorithm against model extraction for the target model (defender) in Section 4.4.

### 4.1 OVERVIEW OF PIPELINE

The GNN model without model reprogramming is demonstrated in Figure 1a. The attack query will still achieve the correct label through the target model since the model is not trained with different objectives for attack and benign queries.

In comparison, the main architecture of the proposed pipeline is shown in Figure 1b. The proposed method shows different processing for queries from different distributions: When the input graph is sampled from the in-distribution data, the model will be modified according to its structural similarity

with the in-distribution data, and the final output label will remain the same with the correct label, thus not affecting the classification utility on benign queries, While the input graph is sampled from OOD data, the model will be reprogrammed, and the output will become different, especially the output labels.

The notations are summarized in Table 2 in Section B in Appendix.

## 4.2 REPROGRAMMING GRAPH MODELS USING DATA-DRIVEN APPROACHES

An ideal goal of defense against ME is to disturb the label attack queries so that the outputs will mislead the clone model, and thus, the clone model will learn modified information about the input graph and labels. However, the reprogrammed model can also affect the results of the benign data. Thus, the reprogramming will affect the result since the parameters of the target model have been changed, resulting in much difficulty when effectively defending against ME.

Confronting these challenges, we propose a pipeline that augments the robustness of GNN by introducing trainable noise into each graph convolution layer. This solution has the potential to significantly bolster the resilience of GNNs against attacks, offering hope for a more secure future in machine learning. For example, in graph property prediction tasks, a regular graph neural network can be classified as message-passing, hidden, and read-out layers, as shown in Eqn. 3. The graph feature can be summarized in $\mathbf{z}_G$.

$$
\begin{cases}
\mathbf{h}_i^{(0)} &= X_i, v_i \in V \\
\mathbf{h}_i^{(l)} &= \sigma\left(\sum_{j \in \mathcal{N}(i)} \frac{1}{c_{ij}} \mathbf{W}^{(l)} \mathbf{h}_j^{(l-1)} + \mathbf{b}^{(l)}\right), v_i \in V, l = 1, 2, \ldots, L \\
\mathbf{z}_G &= \texttt{Readout}(\{h_i^{(L)}, v_i \in V\})
\end{cases}
\tag{3}
$$

In order to let the model reflect different functionalities based on different distributions of the input data for defense, we reprogram the pre-trained target model by injecting layer-wise learnable noise into the output of hidden layers in Eqn. 4,

$$
\mathbf{h}_i^{(l)} = \sigma\left(\sum_{j \in \mathcal{N}(i)} \frac{1}{c_{ij}} \mathbf{W}^{(l)} \mathbf{h}_j^{(l-1)} + \mathbf{b}^{(l)}\right) + \xi^{(l)},
\tag{4}
$$

where the layer-wise noise $\xi^{(l)}$ is learnable noise based on different distributions of the benign queries and can be trained based on the performance of input data, i.e., $\xi \sim q(\cdot|G; \theta_n)$. In practice, the defense performance can update the noise using a gradient descent on Eqn. 6. Thus, the reprogrammed target model is $T_R(x; \theta_T, \xi)$.

Since the target model needs to be trained to perform well on graph classification tasks on benign queries, we can define the task loss for graph inputs with cross-entropy loss to compare the outputs of target models and ground truth.

$$
\mathcal{L}_{\text{task}} = \mathbb{E}_{(x,y) \sim \mathcal{D}_{id}}\left[l_{\text{CE}}(T_R(x; \theta_T, \xi), y)\right],
\tag{5}
$$

By minimizing the task loss $\mathcal{L}_{\text{task}}$, the model can have high classification accuracy on in-distribution benign queries.

Besides, the target model should be reprogrammed with layer-wise noise to evaluate different data distributions. Since a larger distributional difference of model outputs between benign queries and attack queries can be considered a proper attack, we can design the defense loss for training target model:

$$
\mathcal{L}_{\text{defense}} = -\mathbb{E}_{(x,y) \sim \mathcal{D}_{ood}}[D_{\text{KL}}(T_R(x; \theta_T, \xi) \| T(x; \theta_T))],
\tag{6}
$$

By minimizing the loss, the data will maximize the distribution difference between the original pre-trained model and the reprogrammed model on OOD data.

## 4.3 ENHANCE THE DEFENSE WITH GRAPH STRUCTURES

In the model extraction defense pipeline for GNN, the target model not only needs to consider the GNN reprogramming itself, but we also need to consider the data-specific feature of input graphs.

Graph structures can be considered as particular information about distribution. In order to derive the distribution of input graphs, it can be effective to collect graph structure information to identify whether the input graph structure matches the benign queries' graph style. Incorporating graph structure into GNN model extraction defense enhances protection by enabling tailored perturbations that exploit the inherent structural properties of graph data, making it harder for attackers to replicate the model's functionality without degrading its utility for legitimate use.

Thus, we define the normalized graph features $f_G$ as a concatenation of different graph information considering the average degree distribution, clustering coefficient, graph diameter, and spectral features. This can also form a structure distribution of a specific input distribution as $\mathcal{D}_{feat}$. Since the input graph is more similar to the in-distribution graph, it should be more likely to be a benign query. In comparison, lower similarity should be considered a higher probability that the input graph is sampled from $\mathcal{D}_{ood}$. Therefore, we can modify the noise accordingly with the cosine similarity :

$$\alpha = \text{sim}(G_{input}, G_{id}) = \mathbb{E}_{(x,y) \sim \mathcal{D}_{in}} \left[ \cos \angle \left( f_{G_{input}}, f_x \right) \right].$$

where $f_{G_{input}}$ means the structure feature for input graph, and $f_x$ means a selected sample from in-distribution data.

Since the in-distribution graph's structure feature should have a higher similarity with the features selected from training data from the target model, the injected layer-wise noise should be scaled in a lower range. In contrast, the OOD input graph should reprogram the model with a larger scale of layer-wise noise. Details for the structure features and similarity factor are introduced in Section B.3 in Appendix.

## 4.4 Defensive Training Algorithm for ME

Based on the previous analysis, the total loss for the defensive training of the target model against ME can be concluded as follows:

$$
\begin{aligned}
\mathcal{L} =& \mathcal{L}_{\text{task}} + \lambda_1 \mathcal{L}_{\text{defense}} \\
=& \mathbb{E}_{(x,y) \sim \mathcal{D}_{id}} \left[ l_{\text{CE}}(T_R(x; \theta_T, (1-\alpha)/2 \cdot \xi), y) \right] \\
& - \lambda_1 \cdot \mathbb{E}_{(x,y) \sim \mathcal{D}_{ood}} [D_{\text{KL}}(T_R(x; \theta_T, (1-\alpha)/2 \cdot \xi) \parallel T(x; \theta_T))],
\end{aligned}
\tag{7}
$$

which combines the primary task loss with a regularization term for the layer-wise noise parameters $\xi$, thereby enabling the calculation of gradients for the noise alongside the model weights and biases during back propagation.

Reiterating the learning objective, we can confidently conclude the algorithms for training the clone model and the target model. The algorithm for training attackers under the Data-Based Model Extraction (DBME) setting is shown in Algorithm 1. The algorithm includes both hard-label and soft-label settings in model extraction, and the model parameter of the clone model is trained to minimize the output difference with the target model.

The algorithm for training attackers under the Data-Free Model Extraction (DFME) setting is shown in Algorithm 3 in Section B.5 in Appendix. Compared to Algorithm 1, this algorithm includes a graph generator $f_{gen}$ since the clone model can only use synthetic data under the DFME setting. The graph generator is trained in a GAN-like scheme Goodfellow et al. (2020).

Training for defender (target model) is shown in Algorithm 2 in Appendix. In the algorithm, we spend half of the time budget on training the target model with in-distribution data, while the reprogrammed model needs to be trained with both ID data and OOD data with the remaining half-time budget $B/2$.

## 4.5 Theoretical Analysis

In this section, we provide a brief overview of the theoretical justification for the defense ability of our method. Intuitively, the proposed method reduces the attacker's test performance on benign queries by reprogramming the target model to increase the difficulty for clone models.

The key result is summarized in Theorem A.1 in Section A, which states that the proposed objective increases the loss disparity between the clone model and the reprogrammed target model on in-distribution data, effectively lowering the quality of the clone model. Detailed derivations, definitions of distributions, and the proof of Theorem A.1 are provided in the Appendix (Section A.2).

---

**Algorithm 1:** DBME Attack Algorithm

---

**Input:** input graphs batches $\{(x_i)\}$ , pre-trained target model $T$ with parameter $\theta_T$, classifier parameters $\theta_C$
**Output:** Trained clone model $C$ and its parameter $\theta_C$

---

1: Sample input graphs $x_i \sim \mathcal{D}_{ood}$ ;
2: Get target model labels $\hat{y}_i \sim T(x_i; \theta_T)$;
3: Initialize $\theta_C$ for clone model;
4: **for** $(x_i, y_i)$ in training batches **do**
5:     Compute label $\tilde{y}_i = C(x_i, \theta_C)$;
        For *hard-label* settings ;
6:     Compute hard-label loss $\mathcal{L}_Q = l_{\text{CE}}(\hat{y}, \tilde{y})$;
        For *soft-label* settings ;
7:     Compute soft-label loss $\mathcal{L}_Q = l_{\textbf{MSE}}(\hat{y}, \tilde{y})$;
8:     Update $\theta_C$ using $\partial \mathcal{L}_Q / \partial \theta_C$;
9: **end for**

---

Table 1: Clone model accuracy after applying defense methods on **MUTAG** and **ENZYMES** with G_Inception as target model

| Attack | Defense | **MUTAG** Clone Model Architecture | | | **ENZYMES** Clone Model Architecture | | |
|---|---|---|---|---|---|---|---|
| | | GraphSAGE | GIUNET | GIC | GraphSAGE | GIUNET | GIC |
| Soft-label Attack | undefended ↓ | 0.7651 | 0.9342 | 0.9043 | 0.5612 | 0.6855 | 0.6098 |
| | RandP ↓ | 0.7341 | 0.8527 | 0.8351 | 0.5272 | 0.6323 | 0.5726 |
| | P-poison ↓ | 0.7426 | 0.8843 | 0.8756 | 0.5223 | 0.6412 | 0.5813 |
| | GRAD ↓ | 0.7438 | 0.8741 | 0.8321 | 0.5196 | 0.6401 | 0.5627 |
| | AM ↓ | 0.7223 | 0.8661 | 0.8827 | 0.5146 | 0.6335 | 0.5517 |
| | MeCo ↓ | 0.7121 | 0.8234 | 0.8134 | 0.4822 | 0.6197 | 0.5526 |
| | **Ours ↓** | **0.6032** | **0.7829** | **0.7506** | **0.3640** | **0.5754** | **0.5453** |
| Hard-label Attack | undefended ↓ | 0.7346 | 0.8835 | 0.8657 | 0.4874 | 0.6557 | 0.5880 |
| | RandP ↓ | 0.7012 | 0.8087 | 0.7564 | 0.4475 | 0.5821 | 0.5517 |
| | P-poison ↓↓ | 0.7089 | 0.8231 | 0.8054 | 0.4682 | 0.5967 | 0.5458 |
| | GRAD ↓ | 0.7120 | 0.8193 | 0.7901 | 0.4626 | 0.5919 | 0.5530 |
| | AM ↓ | 0.6957 | 0.8125 | 0.7627 | 0.4587 | 0.5830 | 0.5462 |
| | MeCo ↓ | 0.6135 | 0.7974 | 0.7456 | 0.4251 | 0.5724 | 0.5407 |
| | **Ours ↓** | **0.5721** | **0.7531** | **0.7238** | **0.3421** | **0.5447** | **0.5223** |

## 5 EXPERIMENTS

In this section, we evaluate our method and compare it with previous defense methods. We briefly introduce the settings in Section 5.1 and summarize the main results, including defensive performance against different attacks, adaptive attack settings, and ablation studies, in Sections 5.2–5.3.

### 5.1 SETTINGS

We perform experiments on standard graph classification datasets Morris et al. (2020), including **MUTAG**, **ENZYMES**, **NCI1**, and **PROTEINS**. We consider KnockoffNet Orekondy et al. (2019a) as the main query strategy and evaluate our defense against soft-label and hard-label attacks. The target models and defense baselines are briefly described here, with extended details, attacker and defender settings, and dataset partitioning provided in the Appendix (Section C.3.1).

### 5.2 PERFORMANCE

#### 5.2.1 CLONE MODEL ACCURACY

We show the results on four datasets in Table 1 and Table 3 (in Appendix) for the soft-label model extraction attack and hard-label attack setting. The results show that our method can reduce the effectiveness of model extraction methods by up to 17%. The proposed method is much more effective

since: (1) RandP randomly perturbs output possibilities without the data-dependent information, while this may keep the utility for all query data without distributional information misleading; (2) P-poison uses a random surrogate attacker model to work; (3) GRAD: The surrogate model has a large model gap with the ME attacker model; (4) AM applies a distributional detection on input data while it only disturbs the data outputs. (5) MeCo uses the distributional robust training on the model and applies random perturbation based on the distribution, which may not be effective on graph structures.

These results demonstrate that our method performs well compared to recent model defense research against model extraction attacks. Compared to the other information, the layer-wise noise can reprogram the model and lead to misinformation in the output data and the hidden outputs.

### 5.2.2 Target model utility

After applying the defense strategy, we evaluate the model utility by target test accuracy on test ID datasets $D_{id}^{tr}$. We use $l_1$ norm between the reprogrammed target model and the original target model, i.e., $\mathbb{E}_{(x,y) \sim D_{test}} \| T(x, \theta_T) - T^R(x, \theta_T, \xi) \|_1$.

According to the results shown in Table 4 in Appendix, our method can outperform some defense methods in utility. At the same time, the test accuracy on benign data may be lower since the layer-wise noise can affect the model on parameter levels. However, according to the $l_1$-norm, our method can outperform other defense methods. This means our method has a much better ability to output probabilities. The defense baselines have a large $l_1$-norm since they apply perturbation on all queries with the same magnitude. The decrease in test dataset accuracy is a trade-off between dataset performance and preservation of original target model outputs.

Besdies, We also propose an experiment on a large-scale graph dataset, and the results are shown in Table 7 in Appendix. We also compare the defense ability of our method against other graph model extraction attack, and the results are shown in Table 8 in Appendix.

We also briefly evaluate the efficiency and robustness of our method in the Appendix. Our method introduces additional computation for the structure feature extractor and noise generator, which slightly increases inference time compared to simple output perturbation baselines, but remains efficient overall. We also test against adaptive attackers who are aware of our defense strategy; results show that our method effectively misleads clone models, even when attackers adapt their training with partial knowledge of the defense. Detailed measurements and results are provided in the Appendix (Table 6 and 12).

### 5.3 Ablation Studies

We also do the ablation studies of key components in our method, and the details are shown in Section C.6. Specifically, we study the effect of learnable layer-wise noise and graph structure features, the impact of different query budgets for attackers, and hyperparameter choices such as the number of layers adding noise. Our method consistently outperforms baselines under all settings. Detailed results are reported in Table 9 and Table 10 in the Appendix, showing the superiority of learnable noise over random noise, improved defense under limited query budgets, and the influence of noise layer number on clone model accuracy.

## 6 Conclusions

In this paper, we proposed a defense mechanism for Graph Neural Networks (GNNs) against model extraction (ME) attacks, addressing the limitations of existing defenses. Our approach utilizes distributional detection and adaptive reprogramming to protect GNNs without compromising performance on benign queries. By modifying layer outputs and parameters according to input data distributions, we effectively reduce the success of ME attacks, both hard-label and soft-label, while maintaining model utility. Extended experimental results show that our defense strategy significantly weakens the effectiveness of various ME attacks and keeps the defense model's utility. Future work includes improving robustness on diverse graph datasets and optimizing the trade-off between security and utility, while extending the defense to other neural networks and adversarial settings.

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

Table 2: Notations related to data and model

| Variable | Explanation |
|---|---|
| $G$ | Graph |
| $G = (V, E)$ | vertex (node) set $V$ and edge set $E$ |
| $x$ | Graph data in vector and matrix |
| $A$ | Adjacency Matrix, $A \in \mathbb{R}^{n \times n}$, where $n = |V|$ |
| $X$ | Node features matrix, $X \in \mathbb{R}^{n \times d}$, where $d$ is input dimension |
| $y, Y$ | Graph classification label, label set |
| $D$ | $D = \{x_i, y_i\}_{i=1}^N$, graph classification dataset |
| $\mathcal{D}, \mathcal{G}$ | Distribution of (graph) data and labels, $\mathcal{D} = \mathcal{D}_{id} \cup \mathcal{D}_{ood}$ |
| $\mathcal{D}_{id}, \mathcal{G}_{id}$ | In-distribution (ID) graph data, only defender have access to it |
| $\mathcal{D}_{ood}, \mathcal{G}_{ood}$ | Out-of Distribution (OOD) data, attacker trains the clone model with OOD data |
| $D_{id}^{tr}, D_{id}^{te}$ | Train / Test ID dataset |
| $C(\cdot; \theta_C)$ | Clone model |
| $\theta_C, \theta_C^*$ | Model parameter of clone model |
| $T(\cdot; \theta_T)$ | Target model |
| $\theta_T, \theta_T^*$ | Model parameter of target model |
| $v_i$ | The $i^{\text{th}}$ node chosen from node set $V$ |
| $\mathcal{N}(i)$ | The neighbour node set of node $v_i$ |
| $\mathbf{h}_i^{(l)}$ | Hidden layer feature of node $v_i$ in $l^{\text{th}}$ layer |
| $\xi^{(l)}$ | layer-wise noise in the $l^{\text{th}}$ layer |
| $\mathbf{W}^{(l)}, \mathbf{b}^{(l)}$ | Weight and bias in the $l^{\text{th}}$ layer |
| $\mathbf{z}_G$ | Graph feature |
| $T^R(\cdot; \theta_T, \xi)$ | The reprogrammed target model |
| $\alpha$ | Similarity factor |
| $l_{\text{CE}}$ | Cross-Entropy loss for classification |
| $D_{\text{KL}}$ | KL divergence for modeling difference |
| $\mathcal{L}_{\text{task}}$ | Loss to improve classification accuracy |
| $\mathcal{L}_{\text{defense}}$ | Loss to improve defense effectiveness |

# A   DETAILS OF THEORETICAL ANALYSIS

In this section, we propose the theoretical analysis of the defense ability of the proposed method. We will prove that the objective will decrease the clone model's quality on benign queries, which means the proposed method will reduce the attacker's test performance on benign data.

## A.1   DEFENSE ABILITY AGAINST MODEL EXTRACTION

Given two distribution $\mathcal{P}$ an $\mathcal{Q}$ and there probability density function $p(x)$ and $q(x)$, the total variation distance between $P$ and $Q$ is defined as: $\mathbb{TV}(\mathcal{P}, \mathcal{Q}) = \mathbb{E}_x[\|\|p(x) - q(x)\|\|]/2$. The attacker's objective can be denoted by $l(C(x, \theta_C), y)$, which is assumed to be non-negative. The function should be minimized during the training period of the clone model, aiming at increasing the similarity between the output of clone model $C(x, \theta_C)$ and output $y$ (the output of the target model). During the model extraction attack, the attacker finds proper $\theta_C$ to minimize the following objective:

$$\mathbb{E}_{(x,y) \sim \mathcal{D}_{out}}[l(C(x, \theta_C), y)]$$

where $y = T^R(x, \theta_T, \xi)$.

Considering the layer-wise analysis, the attacker minimizes the following objective:

$$\mathcal{L}_{\mathcal{D}_{ood}}(C) = \mathbb{E}_{x \sim \mathcal{D}_{ood}} \mathbb{E}_{\xi \sim q(\xi|x)}[l(C(x, \theta_C), T^R(x, \theta_T, \xi))]. \tag{8}$$

The goal of model extraction for attackers is to reach high test accuracy with clone models on benign queries. To measure the effectiveness of model extraction, we use the disparity in loss between the

clone model and the reprogrammed target model on ID data as:

$$Q(C, T^R) = \mathbb{E}_{(x,y)\sim\mathcal{D}_{in}}[l(C(x, \theta_C), y) - l(T(x, \theta_T), y)],$$

where higher $Q(C, T)$ means worse clone model quality.

We can conclude the following theorem on attackers' quality: Assuming the attacker uses cross-entropy loss $l$, we have

$$Q(C, T^R) \geq \mathbb{E}_{(x,y)\sim\mathcal{D}_{ood}}[\mathbb{E}_{\xi\sim q(\xi|x)}[D_{\mathrm{KL}}(T(x, \theta_T) \parallel T^R(x, \theta_T, \xi))]] - D \tag{9}$$

where $D$ is a fixed constant only based on the target model. The proof of Theorem A.1 is in Section A.2. Theorem A.1 demonstrates that maximizing the KL divergence on OOD data during the model reprogramming training procedure will increase $Q(C, T^R)$, decreasing the quality of the clone model.

## A.2 PROOF OF THEOREM A.1

Theorem A.1 is based on the theoretical analysis in Wang et al..

Suppose $l$ is the loss function. If $\sup l \leq A$, we have

$$Q(C, T) \geq \mathbb{E}_{(x,y)\sim\mathcal{D}_{ood}}[l(C(x, \theta_C), y)] - 4A\mathbb{TV}(\mathcal{D}_{ood}, \mathcal{D}_{id})$$

*Proof.* Firstly, we may consider the difference between ID loss and OOD loss on clone model:

$$|\mathbb{E}_{(x,y)\sim\mathcal{D}_{id}}[l(C(x, \theta_C), y)] - \mathbb{E}_{(x,y)\sim\mathcal{D}_{ood}}[l(C(x, \theta_C), y)]|$$
$$= |\mathbb{E}_{(x,y)}[l(C(x, \theta_C), y)p_{id}(x, y)]$$
$$\quad - \mathbb{E}_{(x,y)}[l(C(x, \theta_C), y)p_{ood}(x, y)]|$$
$$= |\mathbb{E}_{(x,y)}[l(C(x, \theta_C), y) \cdot (p_{id}(x, y) - p_{ood}(x, y))]|$$
$$\leq \mathbb{E}_{(x,y)}[|l(C(x, \theta_C), y)| \cdot |p_{id}(x, y) - p_{ood}(x, y)|]$$
$$\leq |\sup l| \cdot \mathbb{E}_{(x,y)}[|p_{id}(x, y) - p_{ood}(x, y)|]$$
$$= 2A\mathbb{TV}(\mathcal{D}_{ood}, \mathcal{D}_{id}).$$

Similarly, we have

$$|\mathbb{E}_{(x,y)\sim\mathcal{D}_{id}}[l(T(x, \theta_T), y)] - \mathbb{E}_{(x,y)\sim\mathcal{D}_{ood}}[l(T(x, \theta_T), y)]|$$
$$\leq 2A\mathbb{TV}(\mathcal{D}_{ood}, \mathcal{D}_{id}).$$

With the results on target model and clone model, we have

$$Q(C, T) = \mathbb{E}_{(x,y)\sim\mathcal{D}_{id}}[l(C(x, \theta_C), y) - l(T(x, \theta_T), y)]$$
$$\geq \mathbb{E}_{(x,y)\sim\mathcal{D}_{ood}}[l(C(x, \theta_C), y)] - 2A\mathbb{TV}(\mathcal{D}_{ood}, \mathcal{D}_{id})$$
$$\quad - \mathbb{E}_{(x,y)\sim\mathcal{D}_{ood}}[l(T(x, \theta_T), y)] - 2A\mathbb{TV}(\mathcal{D}_{ood}, \mathcal{D}_{id}).$$

Since in our settings, the clone model aims at stealing the functionality of target model on OOD data, we have $y = T(x, \theta_T)$, thus $\mathbb{E}_{(x,y)\sim\mathcal{D}_{ood}}[l(T(x, \theta_T), y)] = 0$. We have

$$Q(C, T) \geq \mathbb{E}_{(x,y)\sim\mathcal{D}_{ood}}[l(C(x, \theta_C), y)] - 4A\mathbb{TV}(\mathcal{D}_{ood}, \mathcal{D}_{id})$$

$\square$

Under our settings with layer-wise noise, the attacker will minimize the objective in Eqn. 8. When we assume the attacker use cross-entropy as learning objective $l$, we have The optimal solution of Eqn. 8 with cross-entropy objective is $\forall x$,

$$C(x, \theta_C) = \mathbb{E}_{\xi\sim q(\xi|x)}[T^R(x, \theta_T, \xi)].$$

*Proof.* Apply the cross-entropy to $l$ in Eqn. 8, we have change the optimization problem to

$$\min_{\theta_C} \mathbb{E}_{(x,y)\sim\mathcal{D}_{ood}}\mathbb{E}_{\xi\sim q(\xi|x)}[-T^R(x, \theta_T, \xi) \cdot \log C(x, \theta_C)],$$

here $\cdot$ means the inner-product of the output vectors.

Assuming the representation ability of the clone model is infinitely large, thus we have

$$\min_{\theta_C} \mathbb{E}_{(x,y)\sim\mathcal{D}_{ood}}\mathbb{E}_{\xi\sim q(\xi|x)}[-T^R(x,\theta_T,\xi) \cdot \log C(x,\theta_C)]$$

$$=\mathbb{E}_{(x,y)\sim\mathcal{D}_{ood}}\left[\min_{\theta_C} \mathbb{E}_{\xi\sim q(\xi|x)}[-T^R(x,\theta_T,\xi) \cdot \log C(x,\theta_C)]\right]$$

$$=\mathbb{E}_{(x,y)\sim\mathcal{D}_{ood}}\left[\min_{\theta_C} -T^R(x,\theta_T,\xi) \cdot \mathbb{E}_{\xi\sim q(\xi|x)}[\log C(x,\theta_C)]\right]$$

To solve the problem, we can derive that the solution of $\theta_C$ is

$$C(x,\theta_C^*) = \mathbb{E}_{\xi\sim q(\xi|x)}[T^R(x,\theta_T,\xi)].$$

This means the clone model can learn the full ability of the reprogrammed target model.  □

Assuming the attacker achieve optimal model extraction solution in Lemma A.2, we have

$$\mathbb{E}_{(x,y)\sim\mathcal{D}_{ood}}[l(C(x,\theta_C),y)]$$
$$\geq\mathbb{E}_{(x,y)\sim\mathcal{D}_{ood}}\mathbb{E}_{\xi\sim q(\xi|x)}D_{KL}(T(x,\theta_T)\|T^R(x,\theta_T,\xi)) - D,$$

where $D$ is a constant only related to target model $T$.

*Proof.* With the optimal solution $C(x,\theta_C^*) = \mathbb{E}_{\xi\sim q(\xi|x)}[T^R(x,\theta_T,\xi)]$ from A.2, We have

$$\mathbb{E}_{(x,y)\sim\mathcal{D}_{ood}}[l(C(x,\theta_C),y)]$$
$$=\mathbb{E}_{(x,y)\sim\mathcal{D}_{ood}}[l(C(x,\theta_C),T(x,\theta_T))]$$
$$=\mathbb{E}_{(x,y)\sim\mathcal{D}_{ood}}[-T(x,\theta_T) \cdot \log C(x,\theta_C)]$$
$$=\mathbb{E}_{(x,y)\sim\mathcal{D}_{ood}}[-T(x,\theta_T) \cdot \log \mathbb{E}_{\xi\sim q(\xi|x)}[T^R(x,\theta_T,\xi)]].$$

To simplify it, we have

$$\mathbb{E}_{(x,y)\sim\mathcal{D}_{ood}}[l(C(x,\theta_C),y)]$$
$$=\mathbb{E}_{(x,y)\sim\mathcal{D}_{ood}}[-T(x,\theta_T) \cdot \log \mathbb{E}_{\xi\sim q(\xi|x)}[T^R(x,\theta_T,\xi)]]$$
$$\overset{(1)}{\geq}\mathbb{E}_{(x,y)\sim\mathcal{D}_{ood}}[-T(x,\theta_T) \cdot \mathbb{E}_{\xi\sim q(\xi|x)}[\log T^R(x,\theta_T,\xi)]]$$
$$=\mathbb{E}_{(x,y)\sim\mathcal{D}_{ood}}\mathbb{E}_{\xi\sim q(\xi|x)}[-T(x,\theta_T) \cdot \log T^R(x,\theta_T,\xi)]$$
$$\overset{(2)}{=}\mathbb{E}_{(x,y)\sim\mathcal{D}_{ood}}[\mathbb{E}_{\xi\sim q(\xi|x)}[D_{KL}(T(x,\theta_T) \| T^R(x,\theta_T,\xi))]] - D$$

where (1) is based on Jensen's inequality and (2) is based on the definition of KL divergence $D_{KL}(P \| Q) := \int p(x)[\log p(x) - \log q(x)]dx$. $D := \mathbb{E}_{(x,y)\sim\mathcal{D}_{ood}}[T(x,\theta_T) \cdot \log T(x,\theta_T)]$ is a constant based on target model , which is fixed during the training procedure of model extraction attack.  □

Combining Lemma A.2 and A.2, we can derive the Theorem A.1.

# B  DETAILS OF METHOD

## B.1  NOTATION TABLE

We summarize the notations for analysis and experiments in this work in Table 2.

## B.2  A SIMPLE DBME SETTING

For DBME, there is a simple class-based division for a simple dataset with multiple labels, e.g., let $Y_{id} = \{0,1,2,3\}$ and let $Y_{ood} = \{4,5,6,7\}$. The distribution separation can be defined as : $\mathcal{D}_{id} = \{(G_i,y_i) \mid y_{id} \in Y_{id}\}$, and we have $\mathcal{G}_{id} = \{G_i \mid y_{id} \in Y_{id}\}$. Of course, $\mathcal{G}_{ood}$ had better to be chosen from other datasets.

### B.3 GRAPH STRUCTURE SIMILARITY

#### B.3.1 GRAPH FEATURE $f_G$

Graph feature for the $f_G$ is based on the train ID dataset $D_{id}^{train}$, which can only be accessed by the defender, which is also the training data for target model. $f_G$ is used to show the graph structure of a group of input graphs. In our experiment, the graph features of $D_{id}^{train}$ is trained to fit the structure, the input is average degree distribution, clustering coefficient, graph diameter, and spectral features. After that, the graph features is trained with GNNs with unsupervised learning Liu et al. (2022); the graph feature of $D_{ood}^{train}$, which is given by the graph neural network is inferenced by the trained model.

#### B.3.2 GRAPH SIMILARITY FOR DISTRIBUTION

While the graph similarity can be computed as the following:

$$\alpha = \text{sim}(G_{input}, G_{id}) = \mathbb{E}_{(x,y)\sim\mathcal{D}_{in}}\left[\cos\angle\left(f_{G_{input}}, f_x\right)\right].$$

where $f_{G_{input}}$ represents the structural feature of the input graph, and $f_x$ denotes a selected sample from the in-distribution data.

In the experiment, we refined $\alpha$ to $\boldsymbol{\alpha} = \{\alpha^{(0)}, \alpha^{(1)}, \ldots, \alpha^{(L)}\}$, which is computed using a normalized inner product applied to the layer-wise noise in each layer of the GNN target model, as the impact factor may vary across different layers. This approach enables the injection of graph structure differences into the model with layer-specific distinctions.

### B.4 ASSUMPTION OF ATTACK QUERY DATA AS OOD DATA

One critical assumption the attacker and defender make is that the attack query data can be treated as out-of-distribution (OOD) data. This assumption is based on the idea that the attacker's queries are typically drawn from a different distribution than the data used to train the target model. By recognizing the attacker's input as OOD, the defender can focus on identifying and mitigating the impact of these queries on the model's performance.

The reason behind this assumption is two-fold: (1) Modern machine learning models (API) require large amounts of labeled data for training, but such data is expensive and typically needs to be fully released in labeled form. This limitation means that attackers cannot access the in-distribution (ID) labeled data for training the target model. As a result, the queries they generate are likely to be OOD concerning the target model's training data. (2) Some data used to train the target model may originate from online users with privacy concerns, preventing the release of data distributions or labels tied to these users. Therefore, without access to this ID data, it is reasonable to assume that all attack queries fall under the category of OOD data.

Thus, we can assume that the attacker can only achieve OOD data to approximate the behavior of the target model.

### B.5 MODEL EXTRACTION ALGORITHM

The Data-based Model Extraction Defense algorithm is shown in Algorithm 2; The Data-Free Model Extraction algorithm for attacker is shown in Algorithm 3.

## C DETAILS OF EXPERIMENTS

### C.1 EXTENDED EXPERIMENTAL DETAILS

#### C.1.1 DATASETS

Detailed descriptions of **MUTAG**, **ENZYMES**, **NCI1**, and **PROTEINS** datasets, including statistics and preprocessing steps.

**Algorithm 2:** Defender Algorithm

**Input:** In-distribution input graphs $x_i$, out-of-distribution graph $x_i'$, model parameters $\theta_T$, layer-wise noise $\xi$, regularization coefficient $\lambda_1$, time budget $B$.
**Output:** Updated model parameters $\theta_T$

1: Sample $x_i \sim \mathcal{D}_{id}$, $x_i' \sim \mathcal{D}_{ood}$ ;
2: Get prediction outputs with $T(x, \theta_T)$ ;
   With half time budget $B/2$ ;
3: **for** $(x_i, y_i)$ in training batches **do**
4:    Update $\theta_T$ with $\mathcal{L}_{task}$ in Eqn. 5 ;
5: **end for**
6: Copy $\theta_T$ to initialize $T^R(x, \theta_T, \xi)$ ;
   With half time budget $B/2$
7: **for** $(x_i, y_i)$ and $(x_i', y_i')$ in training batches **do**
8:    Update $\theta_T$ with $\partial\mathcal{L}/\partial\theta_T$ in Eqn. 7 ;
9:    Update $\xi$ with $\partial\mathcal{L}/\partial\xi$ with Eqn. 7 ;
10: **end for**

---

**Algorithm 3:** DFME Attack Algorithm

**Input:** Input graph $\{x_i\}$, target model $T$ with parameter $\theta_T$, classifier parameters $\theta_C$, data generator $f_{gen}$ parameters $\theta_G$
**Output:** Trained parameters $\theta_C$.

1: Generate input graphs $x_i \sim f_{gen}(\cdot, \theta_G)$ ;
2: Get target model labels $\hat{y}_i = T(x_i, \theta_T)$;
3: Initialize $\theta_C$ for clone model;
4: **for** $(x_i, y_i)$ in training batches **do**
5:    Compute label $\tilde{y}_i = C(x_i, \theta_C)$;
   For *hard-label* settings ;
6:    Compute hard-label loss $\mathcal{L}_Q = D_{\mathrm{KL}}(\hat{y}, \tilde{y})$;
   For *soft-label* settings ;
7:    Compute soft-label loss $\mathcal{L}_Q = l_{\mathbf{MSE}}(\hat{y}, \tilde{y})$;
8:    Update $\theta_C$ using $\mathcal{L}_Q$ ;
9:    Update $\theta_G$ using $\mathcal{L}_G = -\mathcal{L}_Q$;
10: **end for**

### C.1.2 ATTACK AND DEFENSE METHOD

**Query division** We use KnockoffNet Orekondy et al. (2019a) with natural data and describe the specific dataset splits for attack and defense here.

**Attacker settings** We will use two attack methods for training the clone model: (1) **Soft-label** attack, mainly using standard cross-entropy for training the clone model with the probability logits output of the target model and OOD graph data, using soft-label settings; (2) **Hard-label** attack, only the hard-label (top-1 of the output) will be used for training the clone model. We use GraphSAGE Hamilton et al. (2017), GIUNET Amouzad et al. (2024), and GIC Jiang et al. (2019) as different clone models to steal the functionality of the target model.

**Defender settings** As for target models, we use G_Inception Zhao et al. (2018) for MUTAG and ENZYMES and use DUGNN Hayes & Danezis (2018) for NCI1 and PROTEINS. We compared to: (1) Undefended: the target model will not be reprogrammed for OOD data without using any defense strategy; (2) Random Perturb (RandP) Orekondy et al. (2019b): it randomly perturbs the output probabilities of target model; (3) P-poison Orekondy et al. (2019b): it introduces small perturbation to the model's output predictions; (4) GRAD Mazeika et al. (2022): a defense method based on gradient redirection defense; (5) Adaptive Misinformation (AM) Kariyappa & Qureshi (2020): An OOD detection mechanism is combined with the prediction perturbation, only misleading the OOD query data with a misinformation function. (6) MeCo Wang et al. (2023): a recent defense work on

robust model defense against model extraction, we add noise after the first layer of the input graph queries.

**Evaluations**  We mainly evaluate the defense ability through the clone model's test accuracy and the target model's accuracy (utility) on ID data. Besides, we set a large $l_1$ perturbation to 1.0 for the other baselines, that is, $\|y - \widehat{y}\|_1 \leq 1.0$ where $y$ is the modified output probabilities, and $\widehat{y}$ are the unmodified probabilities Wang et al. (2023).

## C.2  SETTINGS

### C.2.1  HARDWARE AND SOFTWARE ENVIRONMENT

We implement models with PyTorch 1.12 and run experiments on a 64-core Ubuntu 20.04 server with NVIDIA GeForce RTX A5000 GPU with 24 GB memories each. It takes 3.5-4 hours to search on a dataset with one million records.

### C.2.2  ATTACKER AND DEFENDER SETTINGS

In our experimental setup, the attacker (clone model) uses an OOD dataset to query the victim model. This OOD dataset differs from the OOD dataset used for reprogramming the target model. The ID data can be divided into a training set and a test set, ensuring that the two sets do not overlap; The train ID dataset can only be accessed by the defender for training target model, and it is not visible to the attacker; The test ID data is used for evaluating the clone quality of the clone model and the utility of the target model; The query data for the attacker uses a different OOD dataset and queries the results from the target model. Then, the attacker uses the query data to train the clone model but tests the clone model on the ID test dataset.

## C.3  EXTENDED PERFORMANCE RESULTS

### C.3.1  PERFORMANCE

Some results of defense performance is shown in Table 3 and Table 4. [1] .

Table 3: Clone model accuracy after applying defense methods on **NCI1** and **PROTEINS** with DUGNN as target model

| Attack | Defense | **NCI1** Clone Model Architecture | | | **PROTEINS** Clone Model Architecture | | |
|---|---|---|---|---|---|---|---|
| | | GraphSAGE | GIUNET | GIC | GraphSAGE | GIUNET | GIC |
| Soft-label Attack | undefended ↓ | 0.6352 | 0.7876 | 0.8207 | 0.7196 | 0.7578 | 0.7523 |
| | RandP ↓ | 0.6034 | 0.7623 | 0.7993 | 0.6788 | 0.6824 | 0.6727 |
| | P-poison ↓ | 0.6036 | 0.7651 | 0.8026 | 0.6861 | 0.7025 | 0.7035 |
| | GRAD ↓ | 0.6055 | 0.7628 | 0.8133 | 0.6801 | 0.7126 | 0.7394 |
| | AM ↓ | 0.6101 | 0.7578 | 0.7981 | 0.6790 | 0.7103 | 0.7125 |
| | MeCo ↓ | 0.5862 | 0.6832 | 0.7363 | 0.6578 | 0.6837 | 0.6737 |
| | **Ours** ↓ | **0.5219** | **0.6564** | **0.6826** | **0.6302** | **0.6538** | **0.6521** |
| Soft-label Attack | undefended | 0.6073 | 0.7425 | 0.7824 | 0.6951 | 0.7164 | 0.7135 |
| | RandP ↓ | 0.5735 | 0.7144 | 0.7459 | 0.6592 | 0.6455 | 0.6820 |
| | P-poison ↓ | 0.5752 | 0.7120 | 0.7634 | 0.6536 | 0.6837 | 0.6852 |
| | GRAD ↓ | 0.5731 | 0.7146 | 0.7661 | 0.6492 | 0.6902 | 0.6813 |
| | AM ↓ | 0.5675 | 0.7235 | 0.7653 | 0.6473 | 0.6923 | 0.6793 |
| | MeCo ↓ | 0.5435 | 0.6946 | 0.6837 | 0.5864 | 0.6771 | 0.6527 |
| | **Ours** ↓ | **0.5024** | **0.6137** | **0.6547** | **0.5003** | **0.6287** | **0.6325** |

---

[1]The code for our experiments can be accessed at `https://anonymous.4open.science/r/GraphModelExtraction-3BEB`.

Table 4: Target model utility (test accuracy) and $l_1$ norm of the output difference

| Defense | MUTAG | | ENZYMES | | NCI1 | | PROTEINS | |
|---|---|---|---|---|---|---|---|---|
| | Accuracy ↑ | $l_1$ norm ↓ | Accuracy ↑ | $l_1$ norm ↓ | Accuracy ↑ | $l_1$ norm ↓ | Accuracy ↑ | $l_1$ norm ↓ |
| undefended | 0.9452 | 0.0 | 0.6629 | 0.0 | 0.8453 | 0.0 | 0.8012 | 0.0 |
| RandP | 0.9250 | 1.0 | 0.6451 | 1.0 | 0.8115 | 1.0 | 0.7864 | 1.0 |
| P-poison | 0.9324 | 1.0 | 0.6324 | 1.0 | 0.8134 | 1.0 | 0.7823 | 1.0 |
| GRAD | 0.9276 | 1.0 | 0.6467 | 1.0 | 0.8072 | 1.0 | 0.7825 | 1.0 |
| AM | **0.9335** | 1.0 | 0.6352 | 1.0 | 0.8025 | 1.0 | **0.7924** | 1.0 |
| MeCo | **0.9297** | 0.3572 | 0.6327 | 0.0649 | 0.8107 | 0.1240 | 0.7871 | 0.1956 |
| **Ours** | 0.9319 | **0.2351** | **0.6467** | **0.0567** | **0.8155** | **0.0762** | 0.7885 | **0.1320** |

Table 5: Clone model accuracy after applying *adaptive attack* on MUTAG with G_Inception as target model

| Attack | Defense | Clone Model Architecture | | |
|---|---|---|---|---|
| | | GraphSAGE | GIUNET | GIC |
| Hard-label Attack | undefended ↓ | 0.7651 | 0.9342 | 0.9043 |
| | **Ours** ↓ | 0.6032 | 0.7829 | 0.7506 |
| | **Ours, Adaptive, unknown architecture** ↓ | **0.5220** | **0.6334** | **0.6101** |
| | **Ours, Adaptive, known architecture** ↓ | 0.5725 | 0.6672 | 0.6502 |
| Soft-label Attack | undefended ↓ | 0.7346 | 0.8835 | 0.8657 |
| | **Ours** ↓ | 0.5721 | 0.7531 | 0.7238 |
| | **Ours, Adaptive, unknown architecture** ↓ | **0.5023** | **0.5942** | **0.5731** |
| | **Ours, Adaptive, known architecture** ↓ | 0.5495 | 0.6247 | 0.5986 |

### C.3.2 INFERENCE TIME

We evaluated the inference time during test time under the same query budget. The results are shown in Table 6, using GraphSAGE as the clone model architecture on the MUTAG dataset. We find that our inference time is higher than that of the undefended model due to the additional layers for the structure feature extractor and noise generator in our method. Compared to RandP, our inference time is still slightly higher because RandP only applies perturbations to the model's output. Compared to AM, our design does not have a surrogate model; thus, it will be less time costly. To summarize, despite the additional processing steps, our method achieves superior inference time efficiency compared to other defense methods.

### C.4 ADAPTIVE ATTACKS

We analyze the robustness of our method against the attacker's adaptive countermeasures. Specifically, we consider the situation where attackers know about our defense and have considered the same model reprogramming scheme when training the clone model without having access to the OOD graph inputs.

According to the results in Table 5, our method can mislead the clone model on the in-distribution data. Besides, the clone models with adaptive attack training perform worse. The reason is that the adaptive attacker can learn the reprogramming scheme while neglecting the robustness of model performance between ID and OOD data.

We also experimented on adaptive attack on data, where the clone model has access to both a subset of in-distribution (ID) data and out-of-distribution (OOD) data for training. From the results in Table 12, we can see that the clone accuracy in the "ID & OOD" setting is higher than in the previous "OOD" setting, as the model can now learn the graph structure from the target model's training data distribution.

### C.4.1 LARGE-SCALE GRAPH TEST

We compared our defense method to a new large-scale dataset for multi-label graph classification, COLLAB, used in social science research. The dataset used for this experiment is a subset of

Table 6: Inference time comparison

| Defense Method | Inference Time (s) |
|----------------|--------------------|
| Undefended | 52.31 |
| RandP | 54.72 |
| P-poison | 432.17 |
| AM | 115.57 |
| **Ours** | 56.42 |

COLLAB, consisting of 3 classes and 1,000 graphs, with an average node count of 74.5 per graph. We conducted a new experiment focusing on model extraction and defense, using GraphSAGE as the clone model architecture. The clone model's accuracy results are as follows: The results in Table 7 show that our method performs effectively on large-scale graph data, maintaining strong model utility while providing robust defense against model extraction.

Table 7: Defense performance on large-scale graph dataset

| Defense Method | Clone Accuracy | Test Accuracy |
|----------------|----------------|---------------|
| Undefended | 0.6978 | 0.6987 |
| RandP | 0.6742 | 0.6389 |
| P-poison | 0.6523 | 0.6420 |
| AM | 0.6426 | 0.6524 |
| **Ours** | **0.6245** | **0.6972** |

### C.4.2 GRAPH MODEL EXTRACTION TEST

We compared our method to different defense methods we use in our work under the GNN-specific attack in Wu et al. (2022a) and Zhuang et al. (2024).

For both of the experiments, we pretrain the target model with architecture of G_Inception on MUTAG. We use GraphSAGE as architecture of clone model, and we evaluate the defense quality on MUTAG. For the attack methods introduced in Wu et al. (2022a), we adapt Attack-2 and Attack-3 to our settings. Specifically, we remove the node attributes from the original graphs, and we use a different subgraph as training and test data for the clone model; For the data-free model extraction attack in StealGNN Zhuang et al. (2024), we adapt Type-III and design a trainable graph generator as inputs for queries.

From these results demonstrated in Table 8, we can see that our method performs well in comparison to other model defense methods against model extraction attacks introduced in Wu et al. (2022a) and Zhuang et al. (2024).

Table 8: Defense performance against graph model extraction attacks

| Defense | Attack-2 | Attack-3 | Type-III |
|---------|----------|----------|----------|
| Undefended | 0.8214 | 0.8430 | 0.7932 |
| RandP | 0.7835 | 0.7921 | 0.7591 |
| P-poison | 0.7576 | 0.7782 | 0.7455 |
| AM | 0.7362 | 0.7495 | 0.7347 |
| Ours | 0.6457 | 0.6649 | 0.6376 |

### C.5 ADAPTIVE ATTACK

### C.5.1 STRONGER ADAPTIVE ATTACK: ARCHITECTURE

The results in Table 5 demonstrate that the adaptive method with a known architecture achieves higher accuracy than the original method and the adaptive attack with an unknown architecture. This is because the target model's architecture is better suited for the task than a randomly selected clone

Table 9: Clone model accuracy on MUTAG with G_Inception as target model

| Attack | Defense | Clone Model Architecture | | |
|---|---|---|---|---|
| | | GraphSAGE | GIUNET | GIC |
| Soft-label Attack | undefended | 0.7651 | 0.9342 | 0.9043 |
| | w/ random noise | 0.6820 | 0.7623 | 0.7409 |
| | w/o structure feature | 0.5732 | 0.6953 | 0.6725 |
| | **Ours** | **0.5220** | **0.6334** | **0.6101** |
| Hard-label Attack | undefended | 0.7346 | 0.8835 | 0.8657 |
| | w/ random noise | 0.6433 | 0.6954 | 0.6793 |
| | w/o structure feature | 0.5421 | 0.6430 | 0.6334 |
| | **Ours** | **0.5023** | **0.5942** | **0.5731** |

Table 10: Sensitivity analysis of layer number on MUTAG with G_Inception as target model

| Attack | Defense | Clone Model Architecture | | |
|---|---|---|---|---|
| | | GraphSAGE | GIUNET | GIC |
| Soft-label Attack | $L_R = 1$ | 0.6725 | 0.6872 | 0.6701 |
| | $L_R = 2$ | 0.5426 | 0.6642 | 0.6532 |
| | $L_R = 3$ | 0.5327 | 0.6723 | 0.6823 |
| | **Ours** | **0.5220** | **0.6334** | **0.6101** |
| Hard-label Attack | $L_R = 1$ | 0.6531 | 0.6726 | 0.6502 |
| | $L_R = 2$ | 0.5312 | 0.6447 | 0.6319 |
| | $L_R = 3$ | 0.5156 | 0.6289 | 0.5920 |
| | **Ours** | **0.5023** | **0.5942** | **0.5731** |

model architecture. Despite this, the results show that our defense method effectively counters even the more advanced adaptive attack.

### C.5.2 STRONGER ADAPTIVE ATTACK: DATA

We have concluded the statement in the introduction: "It is reasonable to assume that all attack queries fall under the category of OOD data," To test this assumption, we conducted an experiment where the clone model has access to both a subset of in-distribution (ID) and out-of-distribution (OOD) data for training. Specifically, we use GraphSAGE as the architecture for the clone model, with a subset of MUTAG (as ID data) and ENZYMES (as OOD data) for model training, where 10% of the data is ID data. We use G_Inception as the architecture for the target model, and we train the target model on the full MUTAG dataset. The clone accuracy results for the previous and new settings under this stronger attack assumption are in Table 12.

From the results in Table 12, we can see that the clone accuracy in the "ID & OOD" setting is higher than in the previous "OOD" setting, as the model can now learn the graph structure from the target model's training data distribution.

Nevertheless, the results demonstrate that our method still provides an effective defense against the updated model extraction attack, even when the clone model has access to both ID and OOD data during training.

### C.6 DETAILS OF ABLATION STUDY

### C.6.1 EFFECT OF MODEL DESIGN

We evaluate the effectiveness of learnable layer-wise noise and graph structure features in our method, and the performance results are shown in Table 9 in Appendix. Our method, when compared to the addition of random noise to different layers, demonstrates a clear superiority. This is due to the fact that random noise, akin to a fix-size model injection, lacks the discerning ability to

Table 11: Query budgets on MUTAG with G_Inception as target model

| Attack | Defense | Query budget (thousand) | | |
|--------|---------|-----------|--------|-----|
| | | GraphSAGE | GIUNET | GIC |
| Soft-label Attack | undefended $\downarrow$ | 213 | 114 | 123 |
| | RandP $\downarrow$ | 245 | 135 | 185 |
| | P-poison $\downarrow$ | 224 | 133 | 135 |
| | AM $\downarrow$ | 264 | 141 | 134 |
| | **Ours** $\downarrow$ | 278 | 170 | 230 |

Table 12: Defense performance on adaptive data attack

| Defense Method | OOD | ID & OOD |
|----------------|--------|----------|
| Undefended | 0.7651 | 0.8174 |
| RandP | 0.7341 | 0.7763 |
| P-poison | 0.7426 | 0.7695 |
| AM | 0.7223 | 0.7570 |
| **Ours** | **0.6032** | **0.6358** |

differentiate between ID and OOD data, a capability our method excels in. Similarly, the graph structure information also contributes to the ability to detect input graphs.

### C.6.2 EFFECT OF DIFFERENT QUERY BUDGETS FOR ATTACKERS

We record the query budgets when the clone model reaches an accuracy of $70\%$ on dataset MUTAG in Table 11 in Section **??**. Our method outperforms various defense methods. This is because the reprogrammed model makes it harder for the clone model to reach high accuracy.

### C.6.3 HYPERPARAMETER CHOICES

One hyperparameter of the target model is the number of layers adding noise $L_R$. In the setting of our methods, it is set to be $L$, which is the layer number of GNNs. While $L_R = 1$, we only add noise into the last layer of the GNNs, when $L_R = 2$, only the last two layers are added layer-wise noise, etc. The results for the layer number analysis are shown in Table 10 . We can derive from the results that the number of layers with noise will affect the clone model accuracy. When $L_R$ grows more significant, the clone model's performance worsens since reprogramming introduces more disturbance into the model.

