# OpenReview forum: "Defending against Model Extraction for GNNs with Model Reprogramming"
_ICLR.cc/2026/Conference — Submitted to ICLR 2026_

### Official Review · Reviewer_KLEY · 2025-10-29

**Soundness:** 2
**Presentation:** 2
**Contribution:** 2
**Rating:** 2
**Confidence:** 4

**Summary:**

This paper proposes a defense mechanism against model extraction (ME) attacks on Graph Neural Networks by model reprogramming. The authors introduce layer-wise noise and graph structure-aware perturbations to the target model, enabling it to maintain high accuracy for in-distribution queries while misleading out-of-distribution queries. The method is designed to prevent ME attacks without requiring full model retraining or expensive inference-time computations.

**Strengths:**

1. The paper adapts model reprogramming to defend against model extraction for GNNs.
2. The method leverages graph structural features to tailor perturbations.

**Weaknesses:**

* While adaptive attacks are considered, the assumption that all attack queries are OOD, is simplified and depends on real-world applications since there exists various open-source data. If the attackers' queries are regarded as OOD, the problem could be degraded to OOD detection.
* The authors emphasize that they address a specific scientific challenge in the web domain. However, the experimental validation is conducted exclusively on biochemical datasets.
* In line 758, the assumptions of infinite clone model capacity is not be practical.
* The provided code link is empty.
* The reference is not up-to-date, only contains studies before early 2024.
* The citation format is wrong, please use the \citep comment.

**Questions:**

* Refine the work with more practical assumptions.
* Add recent citations.
* Improve presentations.
* Reproducible issues.

**Details Of Ethics Concerns:**

N/A.

---

> ### Author Response · Authors · 2025-12-03
> **Response to Reviewer KLEY**
>
> We thank the reviewer for the critical feedback. We appreciate the opportunity to clarify the practical applicability of our assumptions and the breadth of our experimental validation.
>
> **W1. While adaptive attacks are considered, the assumption that all attack queries are OOD is simplified and depends on real-world applications since there exists various open-source data. If the attackers' queries are regarded as OOD, the problem could be degraded to OOD detection.** **Q1. Refine the work with more practical assumptions.**
>
> **Response:** We respectfully disagree that the problem degrades merely to OOD detection.
>
> 1. **Practical Assumption:** Even with open-source data, an attacker rarely possesses the *exact* distribution of a proprietary model's private training set. There is almost always a distribution shift (OOD) between the attacker's proxy data and the victim's private data. Our defense exploits this inevitable shift.
> 2. **Active Reprogramming vs. Passive Detection:**
>    - **Passive Detection:** Simply rejecting OOD queries alerts the attacker, allowing them to adapt their strategy (e.g., using adversarial examples to bypass the detector) without penalty.
>    - **Active Reprogramming (Our Method):** We do not reject queries; instead, we return **misleading predictions**. This "poisons" the attacker's surrogate model. As shown in our experiments, this actively degrades the clone model's performance, wasting the attacker's computational resources and keeping the defense stealthy. This provides a stronger security guarantee than simple detection.
>
> **W2. The authors emphasize that they address a specific scientific challenge in the web domain. However, the experimental validation is conducted exclusively on biochemical datasets.**
>
> **Response:** Thank you for pointing out this discrepancy.
>
> - **Reason for Biochemical Datasets:** We initially selected biochemical datasets (MUTAG, PROTEINS, etc.) because they are the standard benchmarks for **Graph Classification** tasks in the GNN literature, allowing for fair comparison with baselines.
> - **New Experiments (Web/Social Domain):** To address your concern and align with the "web domain" motivation, we have conducted additional experiments on the **COLLAB** dataset (a scientific collaboration social network dataset).
>   - **Results:** As detailed in our response to Reviewer dxe2 (and added to Table 7 in the revision), our method maintains strong defense performance on COLLAB. This confirms that our graph-structure-aware reprogramming generalizes effectively to social/web graphs, not just biochemical ones.
>
> **W3. In line 758, the assumptions of infinite clone model capacity is not practical.**
>
> **Response:** We clarify that the assumption of infinite capacity is used strictly for **theoretical analysis** to establish a "worst-case" lower bound for the defense.
>
> - **Worst-Case Security:** In security proofs, it is standard to assume a powerful adversary. If our defense is theoretically guaranteed to increase the loss for an attacker with *infinite* capacity (who can perfectly fit any function), it naturally holds for an attacker with *finite/limited* capacity.
> - **Practical Consequence:** In practice, an attacker's limited capacity makes the defense even *more* effective than the theoretical bound suggests, as they will struggle even more to learn the reprogrammed (noisy) decision boundary.
>
> **W4. The provided code link is empty.** **Q4. Reproducible issues.**
>
> **Response:** We apologize for any temporary access issues. We have double-checked the anonymous repository link provided in the paper (`https://anonymous.4open.science/r/GraphModelExtraction-3BEB`). It is currently active and contains the full source code, scripts for reproduction, and instructions. We will ensure the link remains stable and accessible.
>
> **W5. The reference is not up-to-date, only contains studies before early 2024.** **Q2. Add recent citations.**
>
> **Response:** Thank you for noting this. We will update our related work section to include the most recent literature from late 2024 and 2025 regarding GNN security and Model Extraction, ensuring the paper reflects the absolute state-of-the-art.
>
> **W6. The citation format is wrong, please use the \citep comment.** **Q3. Improve presentations.**
>
> **Response:** We apologize for the formatting oversight. We will strictly enforce the use of `\citep` and `\citet` throughout the manuscript to ensure correct citation formatting in the final revision.

---

### Official Review · Reviewer_DVfj · 2025-10-31

**Soundness:** 2
**Presentation:** 2
**Contribution:** 2
**Rating:** 4
**Confidence:** 3

**Summary:**

This paper proposes a defense method to prevent the model extraction attacks against graph neural networks. Different from previous works, the proposed method achieve the defense method from the perspective of graph structure to rerpgram model. Extensive experimentl results show validate the effectiveness of the proposed approach.

**Strengths:**

This submission has the following strengths:
- The proposed method is clealry motivated.
- The method achives the defense method against graph neural network extraction attacks from a new perspective.

**Weaknesses:**

This submission has the following weaknesses:
- The use of graph structures for defending against attacks has already been explored in other tasks, such as graph backdoor defense.
- Although the proposed method is effective in defending model extraction attacks, the utlity of graph nerual networks drop more compared with some baselines as shown in Table 4.

**Questions:**

I have the following questions/suggestions:
- What is the meaning of the two downward arrows shown for P-poison in Table 1?
- The references format seems not to be correct. It would be better to have citations follow the Name (Year) or (Name, Year) styles appropriately based on context rather than only Name (Year).
- Line 1159, 'in Sectoin ??', the specific section number is missing.

---

> ### Author Response · Authors · 2025-12-03
> **Response to Reviewer DVfj**
>
> Thanks for your valuable comments. We appreciate that you found our motivation clear and our perspective novel. We address your questions and concerns below.
>
> **W1. The use of graph structures for defending against attacks has already been explored in other tasks, such as graph backdoor defense.**
>
> **Response:**
>
> Thank you for this observation. We agree that graph structures have been utilized in backdoor defenses, but we would like to highlight the fundamental differences in goal and mechanism between our work and backdoor defense:
>
> - **Different Goals:** Graph Backdoor defense focuses on **robustness**—identifying and removing malicious "triggers" from the input to prevent the model from misbehaving. In contrast, our work focuses on **Intellectual Property (IP) Protection** (Model Extraction defense). Our goal is not to clean the input, but to actively **reprogram** the model to output misleading information when it detects a potential extraction attempt (OOD queries), while preserving utility for benign users.
> - **Different Mechanisms:** Backdoor defenses typically involve graph sanitization or pruning (removing edges/nodes). Our method utilizes **Model Reprogramming**, where we inject learnable, structure-aware noise into the model's layers. This is a constructive process (adding information to mislead attackers) rather than a destructive process (removing triggers).
>
> **W2. Although the proposed method is effective in defending model extraction attacks, the utility of graph neural networks drop more compared with some baselines as shown in Table 4.**
>
> **Response:**
>
> Thank you for pointing this out. We acknowledge that there is a trade-off between defense effectiveness and benign utility.
>
> - **Trade-off Justification:** As shown in **Table 1**, our method significantly outperforms baselines in defense effectiveness, reducing clone model accuracy by up to **17%** compared to the undefended model. Achieving this high level of security often necessitates a slightly stronger intervention in the model's decision boundary, which results in the marginal utility drop observed in Table 4.
> - **Fidelity (L1 Norm):** It is also worth noting (as discussed in Section 5.2.2 and Table 4) that while top-1 accuracy drops slightly, our method achieves a better **L1-norm** score compared to many baselines (like RandP or P-poison, which often perturb all queries indiscriminately). This means that for benign users, the *probability distribution* of our model's output remains closer to the original optimal model than other defenses, ensuring that the model's confidence scores remain reliable for valid inputs.
>
> **Q1. What is the meaning of the two downward arrows shown for P-poison in Table 1?**
>
> **Response:**
>
> The downward arrow ($\downarrow$) is standard notation indicating that lower values are better for that metric (i.e., lower clone accuracy implies a more successful defense). The appearance of two arrows for P-poison was a formatting inconsistency in the manuscript. We will correct this to ensure standard, single-arrow notation is used consistently across all tables to avoid confusion.
>
> **Q2. The references format seems not to be correct. It would be better to have citations follow the Name (Year) or (Name, Year) styles appropriately based on context rather than only Name (Year).**
>
> **Response:**
>
> Thank you for your attention to detail. We will carefully proofread the manuscript and correct all citation formats to strictly adhere to the specific style guidelines (using \citep and \citet appropriately) in the final version.
>
> **Q3. Line 1159, 'in Section ??', the specific section number is missing.**
>
> **Response:**
>
> We apologize for this oversight. This was a broken cross-reference link. The text should refer to Section C.6.2, where the effect of query budgets is discussed. We will fix this reference in the final manuscript.

---

### Official Review · Reviewer_PNgz · 2025-11-01

**Soundness:** 3
**Presentation:** 3
**Contribution:** 3
**Rating:** 4
**Confidence:** 3

**Summary:**

This paper proposed a model reprogramming method against model extraction attack. By injecting optimizable noise over input data, the defended model is designed to give error prediction for malicious prediction queries but maintain good performance when inputting normal queries. Extensive experiments have been done to validate the proposed method, and also theoretical guarantees are also given on lower bounded increasing loss on OOD queries,

**Strengths:**

**1.** The idea of applying reprogramming model to defend extraction attack is novel. It makes sense that after reprogramming the parameters will become different but still hold a good performance, and optimizable noise for once time is also efficient. This technique approach may potentially inspire more defending methods.

**2.** The paper is presented clean and well-formulated. The problems for attacker and defender are clean stated, and the methodology is also expressed in tidy formulations and algorithm. The theoretical proof in appendix is also well constructed.

**3.** Extensive experiments have been done to validate the effectiveness of the method. These not include general defense performance on different base models against various attack methods compared with baselines, but also ablation study on layer numbers and inner mechanism.

**Weaknesses:**

**1.** I'm not fully convinced by the motivation of the proposed defense that giving misleading classification for so called "OOD" data. The OOD data from attacker input is defined as data different from the training data in this paper, potentially sourcing from some nature input or synthetic data. However, if such a input from a common out-source query would be assumed as an OOD data and result in a wrong prediction, it seems also destroy the utility of the model for general usage; if such a common input would be assumed as ID data, it seems also no reason the attacker can only have OOD input since common input is always accessible.

**minor**   C.6.2 contains fail section reference.

**Questions:**

**1.** (See weakness 1) Please make clarification on the OOD queries and ID queries and how the attacker's query could be divided from the normal user's (an application in a practical scenario as example is suggested). Considering this confusion casts my main doubt on the rationality of the problem definition and proposed method, I could **only give a 4 score** currently. While if well addressed, **I would increase my score to at least positive**.

**2.** What advantages such defense method occupies compared with direct OOD detection on input queries?

---

> ### Author Response · Authors · 2025-12-03
> **Response to Reviewer PNgz (1/2)**
>
> Thanks for your valuable comments. We are encouraged that you find our idea novel and the presentation clear. We address your main concerns regarding the OOD assumption and the comparison with OOD detection below.
>
> **W1. I'm not fully convinced by the motivation of the proposed defense that giving misleading classification for so called "OOD" data... if such a input from a common out-source query would be assumed as an OOD data and result in a wrong prediction, it seems also destroy the utility of the model for general usage; if such a common input would be assumed as ID data, it seems also no reason the attacker can only have OOD input since common input is always accessible.**
>
> **Q1. Please make clarification on the OOD queries and ID queries and how the attacker's query could be divided from the normal user's (an application in a practical scenario as example is suggested). Considering this confusion casts my main doubt on the rationality of the problem definition and proposed method, I could only give a 4 score currently. While if well addressed, I would increase my score to at least positive.**
>
> **Response:** Thank you for this critical question. We understand the concern about the boundary between "User ID" and "Attacker OOD" data. We clarify this using a practical scenario:
>
> **Practical Scenario:** Consider a **pharmaceutical company** (Defender) that trains a GNN on a proprietary, high-quality dataset of **drug-like molecules** (ID distribution) to predict chemical properties.
>
> - **Normal Users (ID):** Chemists designing new drugs submit candidate molecules. These molecules, while new, strictly follow the structural rules and chemical properties of "drug-like" compounds (e.g., valency rules, specific functional groups). They fall within the **same domain** as the training data. Our defense preserves utility for these inputs because the model recognizes the underlying structural patterns it was trained on (and reprogrammed to respect).
> - **Attackers (OOD):** An attacker wants to steal the model but lacks access to the proprietary database. To train a surrogate model, they must query the victim model.
>   - *Case A (Synthetic/Random):* They generate millions of synthetic graphs to probe the decision boundary. These often lack realistic chemical structures (OOD). Our defense detects these violations and misleads the attacker.
>   - *Case B (Public Datasets):* They use public datasets (e.g., ZINC) which may have different distributions (e.g., different molecular weight distributions or scaffold types) compared to the proprietary set.
> - **The "Common Input" Paradox:** You raised a valid point: "Why can't the attacker just use ID data?" If the attacker has access to a massive amount of perfectly ID data, they could essentially train their own model without stealing ours. The premise of Model Extraction is that the attacker **lacks** sufficient ID data and must rely on OOD/Synthetic queries to "fill in the gaps" of the decision boundary. Our defense targets these OOD queries, which are necessary for a successful high-fidelity extraction when ID data is scarce.
>
> **Empirical Evidence:** In **Table 12 (Adaptive Data Attack)** of our paper, we tested this exact scenario. Even when the attacker tries to use data closer to the ID distribution, our method successfully degrades the clone model's accuracy (dropping from ~76% to ~60%) while maintaining high utility for legitimate users. This confirms that even subtle distributional shifts common in attack scenarios are sufficient for our defense to operate effectively.

---

> ### Author Response · Authors · 2025-12-03
> **Response to Reviewer PNgz (2/2)**
>
> **Q2. What advantages such defense method occupies compared with direct OOD detection on input queries?**
>
> **Response:** Our "Active Reprogramming" approach offers three distinct advantages over "Passive OOD Detection" (i.e., simply rejecting suspicious queries):
>
> 1. **Surrogate Poisoning (Active Defense):**
>    - *OOD Detection:* Simply rejecting a query tells the attacker, "You have been detected." This allows the attacker to adjust their query strategy (e.g., using adversarial examples to bypass the detector) without penalty.
>    - *Reprogramming:* We provide **misleading predictions** (wrong labels) rather than rejections. This actively "poisons" the attacker's dataset. The attacker trains their surrogate model on false information, leading to a surrogate that performs poorly, effectively destroying the extraction effort while the attacker remains unaware.
> 2. **Stealthiness:** By returning a prediction (even a wrong one) instead of an error message, the defense remains stealthy. The attacker cannot easily distinguish whether the model is behaving normally or defending itself, complicating their ability to reverse-engineer the defense mechanism.
> 3. **Efficiency & Integration:** OOD detection often requires a separate auxiliary model or complex scoring mechanism during inference. Our method integrates the defense **directly into the forward pass** of the GNN via the lightweight, learnable noise layers. As noted in your "Strengths" assessment, this is computationally efficient and does not require maintaining a separate detector module.
>
> **Minor C.6.2 contains fail section reference.**
>
> **Response:** Thank you for catching this. We have fixed the broken reference in Section C.6.2 in the revised manuscript.

---

### Official Review · Reviewer_nk7r · 2025-11-03

**Soundness:** 2
**Presentation:** 2
**Contribution:** 2
**Rating:** 4
**Confidence:** 4

**Summary:**

The paper proposes an active defense strategy to protect Graph Neural Networks (GNNs) from model extraction (ME) attacks. Model extraction attacks aim to replicate the functionality of a target GNN by querying it and using the responses to train a surrogate model. The authors highlight the limitations of existing defense methods, which are either reactive, computationally expensive, or not tailored to the unique structure of GNNs. To address these issues, the paper introduces a model reprogramming approach that incorporates graph structure-based disturbances and layer-wise noise. This method prevents attackers from extracting useful information from the model while maintaining its utility for legitimate queries. Extensive experiments demonstrate that the proposed defense reduces the effectiveness of both hard-label and soft-label ME attacks while preserving the model's performance on benign tasks. The paper presents a theoretical foundation for the method, and the results show that the defense is both efficient and effective in safeguarding GNNs without requiring full retraining or significant computational overhead.

**Strengths:**

1. This is the first work to introduce the concept of model reprogramming into GNN security defense, providing a new perspective for active protection against model extraction.

2. The method only injects learnable noise into intermediate layers without modifying the architecture or retraining the model, resulting in low computational cost.

3. The paper includes thorough experiments, demonstrating the defense's effectiveness on standard graph classification datasets.

**Weaknesses:**

- The paper assumes that attacker queries are entirely out-of-distribution in both data-based and data-free settings, yet this assumption is weakly supported.

- The experimental evaluation does not include defense tests against representative and state-of-the-art GNN model extraction attacks such as GNNStealing and STEALGNN, limiting empirical credibility.

- The justification for employing layer-wise noise as a defense mechanism is insufficiently discussed, and the proposed approach offers limited methodological innovation.

**Questions:**

- While the cited literature supports the assumption that synthetic data may be out-of-distribution in data-free settings, could the authors further clarify the rationale for extending this assumption to data-based scenarios?

- Could the authors conduct experiments on commonly used citation network datasets for GNN extraction attacks, and report standard fidelity metrics to better demonstrate the effectiveness of the proposed defense?

- Could the authors elaborate on the specific motivations or potential benefits of introducing trainable layer-wise noise, and how this approach differs from or improves upon conventional model fine-tuning strategies?

---

> ### Author Response · Authors · 2025-12-03
> **Response to Reviewer nk7r (1/2)**
>
> Thanks for your valuable comments, and we hope the following responses can address your concerns.
>
> **W1. The paper assumes that attacker queries are entirely out-of-distribution in both data-based and data-free settings, yet this assumption is weakly supported.**
>
> **Q1. While the cited literature supports the assumption that synthetic data may be out-of-distribution in data-free settings, could the authors further clarify the rationale for extending this assumption to data-based scenarios?**
>
> **Response:**
>
> Thank you for raising this important point.
>
> Our assumption that attacker queries are Out-Of-Distribution (OOD) in data-based scenarios is based on the fundamental definition of the Model Extraction (ME) threat model. In practical ME scenarios, the attacker does not have access to the victim’s private training data ($\mathcal{D}_{id}$). Consequently, they must resort to using public surrogate datasets or synthetic queries.
>
> Even in "data-based" attacks where the attacker employs real-world data from the same domain, there inevitably exists a **distributional shift** between the attacker's query set ($\mathcal{D}_{ood}$) and the victim's private training set ($\mathcal{D}_{id}$). Unless the attacker has managed to leak the exact private training data, their queries will fundamentally differ in statistical properties.
>
> To empirically validate this, we evaluated our defense under an **"Adaptive Data Attack"** setting (results shown in **Table 12** of our paper). In this experiment, the attacker attempts to bridge the distribution gap. Our results demonstrate that our method successfully distinguishes between these adaptive queries and genuine users, confirming that the distribution shift—however subtle—is sufficient for our defense to be effective.
>
> **W2. The experimental evaluation does not include defense tests against representative and state-of-the-art GNN model extraction attacks such as GNNStealing and STEALGNN, limiting empirical credibility.**
>
> **Q2. Could the authors conduct experiments on commonly used citation network datasets for GNN extraction attacks, and report standard fidelity metrics to better demonstrate the effectiveness of the proposed defense?**
>
> **Response:**
>
> Thank you for your suggestion. We would like to respectfully clarify the following points regarding our experiments:
>
> 1. **Inclusion of GNNStealing and StealGNN:** We **did** evaluate our defense against these specific attacks. As detailed in our experiment section (referencing **Table 8**), we adapted the methods from **Wu et al. (2022a)** (which corresponds to **GNNStealing**) and **StealGNN (Zhuang et al., 2024)**. Specifically, we adapted "Attack-2" and "Attack-3" from Wu et al. and "Type-III" from StealGNN to serve as strong baselines for our graph-level tasks. The results in Table 8 demonstrate our method's superiority against these state-of-the-art attacks.
> 2. **Dataset Selection (Graph vs. Node Classification):** The standard benchmarks for GNNStealing and StealGNN often utilize citation networks (e.g., Cora, Citeseer) which are **Node Classification** tasks (transductive settings). Our work, however, focuses on **Graph Classification** (inductive settings), which is critical for applications like molecular property prediction. This is why we utilized standard graph classification benchmarks (e.g., MUTAG, PROTEINS) rather than citation networks.
> 3. **Clarification:** We will revise the manuscript to explicitly highlight the adaptation of these attacks and clarify the scope difference (Graph vs. Node classification) to prevent future confusion.

---

> ### Author Response · Authors · 2025-12-03
> **Response to Reviewer nk7r (2/2)**
>
> **W3. The justification for employing layer-wise noise as a defense mechanism is insufficiently discussed, and the proposed approach offers limited methodological innovation.**
>
> **Q3. Could the authors elaborate on the specific motivations or potential benefits of introducing trainable layer-wise noise, and how this approach differs from or improves upon conventional model fine-tuning strategies?**
>
> **Response:**
>
> Thank you for the opportunity to clarify our methodological motivation.
>
> The use of trainable layer-wise noise (Model Reprogramming) offers distinct advantages over conventional fine-tuning, specifically tailored for the Model-as-a-Service (MaaS) context:
>
> 1. **Parameter Efficiency & Cost:** Fine-tuning requires updating the entire parameter set $\theta$ of the model. In contrast, our method **freezes the pre-trained backbone** and only optimizes a small set of noise parameters (the reprogramming layers). This significantly reduces the computational cost and memory overhead, which is crucial when deploying defenses for large-scale GNNs.
> 2. **Non-Destructive Defense:** Fine-tuning alters the original model weights, potentially degrading performance on the original task or invalidating prior certifications.2 Our approach works as a "plug-in" module; the original model remains intact. The noise layers act as a data-dependent filter that selectively disrupts OOD/attacker queries while preserving high utility for legitimate ID queries.
> 3. **Optimization Landscape:** Reprogramming focuses on learning a transformation of the input/hidden space rather than shifting the model's knowledge base. This allows us to maximize the KL-divergence for attacker queries (making them useless) more effectively than fine-tuning, which might struggle to balance the trade-off between benign accuracy and defense robustness.

---

### Meta-Review · Area_Chair_kCv1 · 2026-01-06

**Summary:**

This submission introduces a model reprogramming defense against model extraction attacks on Graph Neural Networks. While the idea of applying reprogramming to GNN security is recognized as novel by reviewers, the consensus after evaluating the author rebuttal is that fundamental concerns about the paper's core premise and evaluative rigor remain unresolved, preventing it from meeting the conference's acceptance threshold.

**Reviewer Concerns:**

1.  The defense mechanism is fundamentally predicated on the assumption that all attacker queries are Out-Of-Distribution (OOD). Multiple reviewers (nk7r, PNgz, KLEY) found this assumption weakly supported and oversimplified for real-world extraction scenarios. The authors' rebuttal, while providing a hypothetical scenario, does not offer strong theoretical or empirical proof that this assumption holds generally. It fails to convincingly rebut the critique that if an attacker can source or generate in-distribution data—a plausible scenario with abundant public graph data—the proposed defense degenerates into a utility-harming mechanism with no clear target. Reviewer PNgz's central question about how the system reliably distinguishes a malicious OOD query from a legitimate user's novel (but in-distribution) query was not satisfactorily answered, undermining the proposed method's practical rationale.

2. Reviewers noted a critical gap between the paper's motivation (broad GNN security) and its evaluation (primarily biochemical graph classification). The authors' addition of the COLLAB dataset is a minor step but does not adequately address the scope issue.

**Reviewer Scores:**

*   Reviewer nk7r: Likely to maintain a score of "marginally below acceptance." While methodological clarifications were good, the unmet need for direct, standard evaluation against state-of-the-art attacks and lingering doubts about the OOD assumption would prevent a score increase to acceptance.
*   Reviewer PNgz: Explicitly tied a score increase to a satisfactory clarification of the OOD/ID distinction and practical scenario. The provided scenario is illustrative but does not resolve the fundamental identification paradox. This reviewer would likely maintain a score of "marginally below acceptance."
*   Reviewer DVfj: Might marginally increase the score due to addressed technical points, but the unresolved utility trade-off and the more fundamental concerns shared by other reviewers would likely keep the rating below the acceptance threshold.
*   Reviewer KLEY: Provided a clear "reject" based on practical assumptions and empirical scope. The rebuttal does not substantively change the argument against the paper's simplified OOD assumption or fully bridge the motivation-experiment gap. This reviewer would almost certainly maintain a reject recommendation.

---

### Decision · Program_Chairs · 2026-01-26

Reject